# Investigating Whether Bereavement by Suicide and Other Unnatural Causes Has a Deterrent Effect on Alcohol or Drug Use in Young Adults

**DOI:** 10.3390/ijerph192013245

**Published:** 2022-10-14

**Authors:** Alexandra Pitman, Fiona Stevenson, David Osborn

**Affiliations:** 1Division of Psychiatry, University College London (UCL), London W1T 7NF, UK; 2Camden and Islington NHS Foundation Trust, London NW1 0PE, UK; 3Department of Primary Care and Population Health, University College London (UCL), London NW3 2PF, UK

**Keywords:** bereavement, grief, alcohol, drugs, suicide, cause of death, mixed methods

## Abstract

Bereavement by suicide and other unnatural causes is associated with suicide but evidence regarding risk of substance misuse is inconsistent. This may be due to heterogeneity in patterns of alcohol or drug use after traumatic bereavement; some increasing use to cope with the loss and others reducing use. To highlight the problems of focussing on diagnostic thresholds when investigating substance use after traumatic loss, we aimed to test whether people bereaved by suicide or other unnatural causes are more likely to reduce or stop their substance use than people bereaved by sudden natural causes. Using multivariable logistic regression and data from an online survey of 1854 UK-based bereaved adults, we tested the association between bereavement by suicide and other unnatural causes and post-bereavement reduction/cessation in (i) alcohol and (ii) drug use. There were no group differences in the proportions who reduced/stopped alcohol use, but a significantly greater proportion of people bereaved by sudden unnatural causes reduced/stopped drug use post-bereavement than people bereaved by sudden natural causes (AOR = 2.61; 95% CI = 1.44–4.71; *p* = 0.001; 4.1% versus 1.7%). In sub-group analyses this applied separately to people bereaved by suicide and non-suicide unnatural causes. Research into post-bereavement substance use should accommodate apparent divergent sub-diagnostic patterns.

## 1. Background

Although bereavement is a near universal experience, and grieving is the natural process by which people adapt to a loss, for some individuals adjustment can be more difficult, with substantial variation in psychological health between individuals and across cultures [1]. Bereavement by any cause is associated with an excess risk of physical and mental health problems [1], suicide [2] and premature all-cause mortality [1]. However, people who experience bereavement due to unnatural causes (such as suicide, homicide or accidental death) are at greater risk of psychiatric illness and suicide than those bereaved by other causes [3]. These types of deaths are also more stigmatised than deaths by natural causes [3]. In attempting to understand mechanisms of suicide risk after bereavement by unnatural causes, researchers have investigated substance use, identifying this as a likely mediator because it is a common means of processing bereavement [4] and a potent risk factor for suicide [5]. This work has tended to rely on recorded diagnoses of substance use disorder or diagnostic interviews as outcomes, finding elevated risks of substance misuse after bereavement by suicide and other unnatural causes when compared with non-bereaved controls [6,7,8], but no differences when comparing suicide bereavement to bereavement by other unnatural causes [7,9]. However, such work ignores patterns of substance misuse at a sub-diagnostic level, hampering a more nuanced understanding of whether individuals respond differently in their use of substances after a traumatic loss. In the current study, prompted by qualitative findings of varied and divergent patterns of substance misuse after traumatic loss [10,11,12,13,14], we aimed to test a contrary hypothesis using more fine-grained measures of substance use. We hypothesised that people bereaved by unnatural causes (including suicide) are more likely than those bereaved by natural causes to reduce or stop their use of alcohol and drugs after bereavement. This has not yet been investigated, and was intended to complement findings that these groups are also more likely to increase their use of substances (alcohol or drugs) after a bereavement [12], improving our clinical understanding of potentially differing patterns within this population and making a methodological point about factors to consider when investigating substance use after bereavement.

## 2. Materials and Methods

### 2.1. Sample

We analysed cross-sectional data from adults who had participated in the 2010 UCL Bereavement Study; a closed online survey of adults aged 18 to 40 working or studying at 37 British higher education institutions (HEIs) who were invited to take part in a survey of “*the impact of sudden bereavement on young adults*”(see Appendix A) [15]. All 164 HEIs in the UK at that time had been invited to participate, and 37 had agreed. Each staff and student member of the 37 HEIs (an estimated sample of 659,572 in a sampling frame of 20% of all 164 UK HEIs) received an individual email inviting them to participate in a survey investigating the impact of sudden bereavement on young adults and defining sudden bereavement as “*a death that could not have been predicted at that time and which occurred suddenly or within a matter of days*”. This allowed identification of bereaved individuals without using a help-seeking sample, and was judged to be the optimal means of recruiting a hard-to-reach population of young adults with broad socio-economic and geographic representation [15].

Respondents self-identified type of bereavement as: bereavement by suicide, bereavement by sudden natural causes (for example cardiac arrest), and bereavement by sudden unnatural causes (for example accidental death). In the case of exposure to more than one mode of sudden bereavement, all those bereaved by suicide were asked to relate their responses to their bereavement by suicide (and in the case of more than one suicide bereavement, to the person they had felt closest to). Those bereaved by deaths due to sudden natural causes and to sudden unnatural causes were asked to relate their responses to whichever person they had felt closest to, identifying exposure status accordingly.

### 2.2. Procedures

Participants completed an online survey that had been designed and piloted in consultation with a group of young, bereaved adults and bereavement counsellors, capturing quantitative and qualitative data on important domains in relation to the impact of bereavement. As well as eliciting quantitative data on socio-demographic and clinical characteristics [15], open questions elicited free-text qualitative data on issues such as the impact of the bereavement on relationships [16], occupational functioning [17], and support needs [18]. Respondents were invited to give as much or little detail as they wished, with no upper word limit, or to skip the question if it did not apply. The question relating to use of drugs and alcohol was worded as follows: “*In what way, if any, has the bereavement affected your drinking habits or your use of unprescribed drugs? (Unprescribed drugs include illicit drugs as well as medications used above their prescribed limits)”.* The wording of this question was designed to be non-leading, neutral (avoiding assuming solely negative outcomes of bereavement), and unambiguous in using the term unprescribed drugs to cover use of illegal drugs, legal highs, over-the-counter drugs, or prescribed drugs used above advised limits.

### 2.3. Ethical Approval

The UCL Bereavement Study was approved by the UCL Research Ethics Committee (ref: 1975/002). All participants provided online informed consent by ticking a box to indicate they had read the participant information leaflet and consent form and agreed that the anonymised results would be used for research purposes.

### 2.4. Measures

#### 2.4.1. Outcomes

As reported in our previous mixed methods study describing substance use after sudden bereavement [12], we used the approach of content analysis to analyse large volumes of brief free-text responses to the question on drug and alcohol use to capture perceptions of any increases, decreases, or no changes. This broad approach to categorisation avoided attempting diagnostic categorisation or estimates of the quantity of substances used. Text was analysed with the researchers blinded to the cause of death, except for unblinding in 15 cases where the text mentioned cause.

Two authors (AP; FS) conducted content analysis collaboratively with team discussions providing opportunities to check the validity of codes against data, clarify where meaning was uncertain, and encourage reflexivity. Our cross-disciplinary team comprised clinical and non-clinical perspectives to challenge differences in interpretation.

Our initial content analysis identified 11 mutually exclusive categories capturing the impact of the bereavement on substance use: (i) no change (including those who were abstinent pre-and post-bereavement); (ii) stopped; (iii) reduced; (iv) brief temporary increase (within the week of the death) but then resumed pre-loss pattern of use; (v) increased (unclear if perceived as helpful or harmful); (vi) increased (perceived as helpful); (vii) increased (perceived as harmful); (viii) increased (unclear if perceived as helpful or harmful) but then resumed pre-loss pattern of use; (ix) increased (perceived as helpful) but then resumed pre-loss pattern of use; (x) increased (unclear if perceived as helpful or harmful) but then stopped; and (xi) unable to classify.

We used these categories to create two binary variables: reduction/cessation in alcohol use post-bereavement, and reduction/cessation in drug use post-bereavement. Each used the categories *stopped* and *reduced* above to define a positive outcome specific to alcohol or drugs, and the other nine categories to define a negative outcome.

#### 2.4.2. Exposure

We derived a binary exposure variable to compare all those who related their survey responses to (a) bereavement by suicide or by other sudden unnatural causes and (b) bereavement by sudden natural causes.

#### 2.4.3. Covariates

We chose six variables *a priori* as potential confounders based on previous literature [3] and clinical judgement. Survey data captured these variables as follows:Age: continuous measure, defined by participant (options from 18–40 years)Gender: binary variable, defined by participant (male; female)Time since bereavement: continuous measure, defined by participant (years/months)Socio-economic status: categorical measure, derived from a question capturing own occupation (for HEI staff) or parental occupation (for students), using the 5 categories used by the Office for National Statistics (ONS)Pre-bereavemen: depression: binary measure, using the Composite International Diagnostic Interview (CIDI) screen for lifetime depression, qualified by whether this was before or after the sudden bereavement, to derive a pre-exposure measureFamily history of psychiatric problems (including drug and alcohol problems): binary measure derived from responses to the question “Has anyone in your family suffered from an anxiety disorder, a depressive disorder (including postnatal depression), had drug or alcohol problems, or other psychological or emotional difficulties?”

### 2.5. Statistical Analysis

We described the socio-demographic and clinical characteristics of our sample using χ^2^ tests (categorical variables) and one-way analysis of variance (continuous variables) to compare people bereaved by sudden unnatural causes and sudden natural causes.

We used multivariable logistic regression to test the association between bereavement by unnatural causes and our two outcomes (post-bereavement reduction/cessation in alcohol use; post-bereavement reduction/cessation in drug use) adjusted for six potential confounders as listed above. We fitted binary models using *xtlogit* commands in Stata, taking into account any clustering effect at institutional level by estimating random effects for each cluster (n = 37 HEIs).

To identify whether findings differed by whether an individual was bereaved by suicide or non-suicide unnatural causes, we conducted sub-group analyses in which we ran the above models comparing each of these groups separately to the control group of people bereaved by sudden natural causes.

We conducted all analyses in Stata version 16 and used a 2-sided *p*-value threshold of <0.05 for all models.

## 3. Results

### 3.1. Response

Of the 5085 bereaved by the sudden death of a close contact who responded to the UCL Bereavement Study questionnaire, 91% eligible adults (n = 4630) consented to participate, and 1854 (40%) responded to the open question on substance use and were therefore included in the current analysis. There was no accurate way of measuring overall response to the survey, as the denominator of bereaved people could not be ascertained.

Of the 1854 adults who provided free-text data on substance use, 353 were bereaved by suicide, 395 by sudden unnatural causes and 1106 by sudden natural causes. We therefore compared the 748 adults bereaved by unnatural causes (353 by suicide; 395 by sudden unnatural causes) to 1106 people bereaved by sudden natural causes.

### 3.2. Sample Characteristics

The overall gender balance of the sample was 19% male and 81% female, with a mean age of 25.6 years (SD = 6.3). People bereaved by sudden natural causes were significantly more likely than those bereaved by unnatural causes to have been under 21 years when responding, to have been bereaved more recently, and to have been related to the deceased (Table 1).

In the full sample 7.7% reduced/stopped their use of alcohol post-bereavement and 2.7% reduced/stopped their use of drugs, whilst alcohol use was unchanged for 58% and drug use was unchanged for 85% (including those who had never used alcohol/drugs).

### 3.3. Association of Bereavement Status with Reduction/Cessation in Substance Use

There were no significant group differences in the proportions who had reduced or stopped their alcohol use (Table 2). A significantly greater proportion of people bereaved by sudden unnatural causes reduced/stopped their use of drugs after the bereavement (AOR = 2.61; 95% CI = 1.44–4.71; *p* = 0.001; 4.1% versus 1.7%) than people bereaved by sudden natural causes.

In sub-group analyses (Table 3), compared with people bereaved by sudden natural causes, significantly greater proportions of people bereaved by suicide (AOR = 3.13; 95% CI = 1.60–6.13; *p* = 0.001; 5.1% versus 1.7%) and of people bereaved by non-suicide unnatural causes (AOR = 2.19; 95% CI = 1.06–4.53; *p* = 0.035; 3.3% versus 1.7%) had reduced/stopped their use of drugs post-bereavement, but with no group differences for alcohol use (7.1% versus 8.5% versus 8.6%, respectively).

## 4. Discussion

### 4.1. Main Findings

We found that it was more common for people bereaved by sudden unnatural causes (including suicide) to reduce or stop their use of illicit drugs after the loss than people bereaved by sudden natural causes. However, we found no differences in the proportions reducing or stopping their alcohol use. Our sub-group analyses showed that the greater tendency to reduce or stop illicit drug use applied individually to people bereaved by suicide and people bereaved by sudden unnatural causes excluding suicide, whilst acknowledging small numbers and limited power. These findings were in the context of the small minority of people in the overall bereaved sample who reduced or stopped their use of alcohol (8%) or drugs (3%) after the loss.

Our differing findings for drugs and for alcohol could reflect a perception that drugs pose a greater risk to health than alcohol. Alcohol is regarded as a socially acceptable way to cope with adversity in many Western societies, with social and cultural influences promoting the practice of self-medicating with alcohol when coping with life stressors such as bereavement [4]. Such influences may be particularly pronounced among young British people [19]. Given the predominance of females in our sample, our findings for drug use may reflect the impact of losing a drug-using partner as well as their facilitative influence on drug use at home or in leisure spaces [20,21].

Another explanation for these findings is that social support is protective against substance misuse [4], and that people bereaved by unnatural causes receive a level of social support that reduces their reliance on drugs. However, this is contradicted by evidence describing the higher levels of stigma perceived by people bereaved by unnatural causes [3] and their lower levels of support [22], particularly for those bereaved by suicide [22,23]. Individuals who reduce their use of drugs after a traumatic bereavement warrant close study to understand how they achieve this in the context of poor social support, and whether their beliefs about the potential harms of drug use are rooted in heightened health anxiety.

### 4.2. Findings in the Context of Other Studies

No other studies have tested this hypothesis, and our findings complement those of our previous study showing that people bereaved by suicide and people bereaved by non-suicide unnatural causes are both more likely to increase their use of substances after bereavement than people bereaved by natural causes (when considering alcohol or drugs together) [12]. However, when considering alcohol and drugs separately, only the group bereaved by non-suicide unnatural causes are more likely to increase their use of alcohol compared with those bereaved by natural causes [12]. Together, this suggests that people bereaved by non-suicide unnatural causes are more likely to reduce or stop their drug use and also more likely to increase their alcohol use. Our findings regarding no group differences in alcohol reduction/cessation are consistent with quantitative work using Danish registers, which found an elevated risk of alcohol or drug use disorder in suicide-bereaved partners compared with non-bereaved partners, but no differences between suicide-bereaved partners and other-bereaved controls [7]. However, our findings of no differences in alcohol reduction/cessation are inconsistent with findings from a Danish analysis showing that suicide-bereaved partners have a reduced risk of liver cirrhosis (an alcohol misuse marker) when compared to non-bereaved partners and to partners bereaved by other causes [7]. It is possible that such varied findings are interpretable in the context of divergent patterns within the bereaved population, such that patterns differ for those with a formal diagnosis of substance misuse and for those consuming at sub-diagnostic levels. Our study did not investigate sex or age differences due to limited statistical power, but previous work shows that after suicide loss women may be more likely to report using prescription drugs, men may be more likely to report illicit drug use and alcohol use, and younger people may be more likely to report substance misuse [24].

Whilst the more common narrative is that of using alcohol after bereavement to dull the pain and purge sadness [25] the findings of the current study identify a group who find alternative ways of coping. Our findings are consistent with the qualitative accounts of people bereaved by suicide [11] and non-suicide unnatural causes [10] who describe conscious efforts to restrain their use, in order to help them cope. Reasons given include an awareness that alcohol or drugs lower mood, hamper control of emotions, or increase fears that they could become like the person who died [10,11,12]. People bereaved by alcohol-related deaths describe being confronted by their own mortality and realising that their own substance misuse may pose a risk to their health [26]. They also describe wanting to learn more about substance misuse to better understand the deceased and their problems [27]. Such accounts do not distinguish clearly between attitudes towards drugs and attitudes towards alcohol, and therefore how this might apply differentially to consumption of each. More work is needed to understand the cognitions of those who change their use of alcohol or drugs after traumatic loss, or order to help design interventions that promote coping and recovery.

### 4.3. Strengths and Limitations

We analysed data from what we believe to be the largest-scale study collecting qualitative data on self-reported use of alcohol and unprescribed drugs after different modes of sudden bereavement. Nevertheless, low event rates in our sample of 1854 bereaved adults meant we had limited power for statistical models. Recruitment avoided use of a help-seeking sample, and outcomes did not rely on narrow diagnostic criteria. Collection of fine-grained data on consumption of drugs and alcohol in the day-to-day lives of respondents captured more nuanced changes in patterns of drinking and drug use after a negative life event. Registry-based studies using recorded diagnoses of substance misuse or dependence do not permit such a detailed investigation of reduced intake or of reported harmful use. The anonymous format of data collection also promoted disclosure, although did not allow for further probing. Our cross-disciplinary team approach encouraged personal reflexivity when analysing free-text data by challenging differences in interpretation, reducing the influence of theoretical or personal conceptions [28]. However, we acknowledge that subjectivity in coding, social desirability bias, and recall bias may have over-ascertained (particularly in the context of social desirability bias) or under-ascertained our outcomes, biasing our quantitative findings. Our multi-level models were adjusted for covariates agreed *a priori*, although we acknowledge the possibility of residual confounding. Comparison to sudden natural bereavement rather than any bereavement by natural causes took into account the sudden or unexpected nature of the death, which might also be a factor in promoting use of alcohol or drugs. As females of all ages have a greater risk of bereavement [29] we would expect an excess of bereaved females aged 40 and below, particularly given the much higher suicide and accidental death rate in males than females in this age group [30]. However, our findings from a predominantly female (81%), white (90%), high socio-economic status (60%) sample may not be generalisable to all young bereaved adults in the UK or internationally. Given that drug-related deaths of peers and partners are likely to be over-represented in this younger sample, our findings may not be generalisable to other age groups.

### 4.4. Clinical and Research Implications

Our findings demonstrate that clinicians cannot assume uniform responses to a traumatic bereavement and that a small minority of bereaved people make positive changes to lifestyle factors. The extent that this is driven by health anxiety remains unclear. Whilst other work (analysing the same dataset) shows that a significantly greater proportion of suicide-bereaved individuals increase their use of drug or alcohol use after a sudden loss [12], the current study shows that another sub-group appear motivated to reduce their risk of misusing unprescribed drugs. How these two groups differ in their adjustment to the loss is unclear and requires longitudinal investigation. Those who reduce their use of alcohol or drugs are of interest for their coping strategies. Those who increase their use are concerning because they may be those in most distress and those at greatest risk of suicide attempt. Further longitudinal work is needed to understand the socio-demographic and cognitive characteristics and health outcomes of each group. This will help understand the role of alcohol and drug use in the association between suicide bereavement and suicide.

Patterns of use tend to change over the course of grief. Cohort studies should pay particular attention to the reported tendency for some people bereaved by suicide to increase their use of alcohol or drugs in the early stages of the loss [13] but to reduce this over the next two years [14], as also observed in samples of people bereaved by sudden natural and unnatural causes [12]. Clinicians should be aware of the potential for bereaved individuals to react in a range of ways after a loss, and for these patterns to change over the course of grief. A sensitive discussion about the perceived benefits and risks of drug and alcohol use is important given that some bereaved individuals report finding alcohol helpful in coping with grief over the short-term [10,11,25]. Where an individual is identified as being at risk, educational resources and motivational approaches may be helpful in considering appropriate responses, building on the reported awareness of the potential for negative effects in longer-term use [25]. Future research should explore the predictors of increased and reduced alcohol and drug use after a traumatic life event, and use these to screen sensitively for hazardous use.

## 5. Conclusions

A small minority of people bereaved by sudden unnatural causes report a reduction in their use of drugs or alcohol after the bereavement, but for drug use this proportion is significantly greater than that for people bereaved by sudden natural causes. This applies both to people bereaved by suicide and to people bereaved by non-suicide unnatural causes when compared to people bereaved by sudden natural causes. It is possible that experiencing the death of a close contact by unnatural causes can influence drug use attitudes and behaviour of some individuals who seek to protect their own health. Further work is needed to understand the cognitions associated with this behaviour, and whether this has a buffering influence on the adverse health outcomes of bereavement by sudden unnatural causes.

## Figures and Tables

**Table 1 ijerph-19-13245-t001:** Sociodemographic and clinical characteristics of bereaved sample.

Bereavement Exposure	Sudden Natural Causes (n = 1106)	Sudden Unnatural Causes (Suicide and Non-Suicide Unnatural Deaths) (n = 748)	Sub-Groups of the n = 748 Bereaved by Sudden Unnatural Causes		Total (n = 1854)	p-Value †
			Suicide (n = 353)	Sudden (Non-Suicide) Unnatural Causes (n = 395)		
**Socio-demographic characteristics**	**n (%)**	**n (%)**	**n (%)**	**n (%)**	**n (%)**	
**Gender ^††^**						0.397
male	217 (19.6)	135 (18.1)	67 (19)	68 (17.2)	352 (19.0)	
female	889 (80.4)	613 (82.0)	286 (81)	286 (81.0)	1502 (81.0)	
missing	0 (0)	0 (0)	0 (0)	0 (0)	0 (0)	
**Age of participant (binary variable) ^a^**						**0.047**
aged 18–21	412 (37.3)	245 (32.3)	112 (31.7)	133 (33.7)	657 (35.4)	
aged 22–40	694 (62.8)	503 (67.3)	241 (68.3)	262 (66.3)	1197 (64.6)	
missing	0 (0)	0 (0)	0 (0)	0 (0)	0 (0)	
**Age of participant (years) ^††^**						0.986
mean (SD)	25.5 (6.4)	25.7 (6.2)	25.8 (6.3)	25.6 (6.0)	25.6 (6.3)	
**Age participant was bereaved**						0.761
between age 10 and 17	440 (39.8)	303 (40.5)	143 (40.5)	160 (40.5)	743 (40.1)	
between age 18 and 40	664 (60.0)	444 (59.4)	209 (59.2)	235 (59.5)	1108 (59.8)	
missing	2 (0.2)	1 (0.1)	1 (0.3)	0 (0)	3 (0.2)	
**Self-defined ethnicity**						0.186
white	994 (89.9)	685 (91.6)	323 (91.6)	362 (91.7)	1679 (90.6)	
non-white	112 (10.1)	62 (8.3)	30 (8.5)	32 (8.1)	174 (9.4)	
missing	0 (0)	1 (0.1)	0 (0)	1 (0.3)	1 (0.1)	
**Socio-economic status ^††,b^**						0.365
social classes 1.1 & 1.2	680 (61.5)	474 (63.4)	222 (62.9)	252 (63.8)	1154 (62.2)	
social classes 3–7 & 9	402 (36.4)	256 (34.2)	124 (35.1)	132 (33.4)	658 (35.5)	
missing	24 (2.2)	18 (2.4)	7 (2.0)	11 (2.8)	42 (2.3)	
**Clinical characteristics**						
**Pre-bereavement depression ^††,c^**						0.540
Yes	230 (20.8)	164 (21.9)	92 (26.1)	72 (18.2)	394 (21.3)	
No	876 (79.2)	582 (77.8)	260 (81.5)	322 (81.5)	1458 (78.6)	
missing	0 (<0.1)	2 (0.3)	1 (0.3)	1 (0.3)	2 (0.1)	
**Family history of any psychiatric problems** (including drug and alcohol problems) ^††^						0.104
Yes	737 (66.6)	525 (70.2)	255 (72.2)	270 (68.3)	1262 (68.1)	
No	368 (33.3)	222 (29.7)	98 (27.8)	124 (31.1)	590 (31.8)	
missing	1 (0.1)	1 (0.1)	0 (0)	1 (0.3)	2 (0.1)	
**Characteristics of the bereavement**						
**Years since bereavement ^††,a^**						**0.002**
less than two years	376 (34.0)	204 (27.3)	98 (27.8)	106 (26.8)	580 (31.2)	
two years or more	730 (66.0)	544 (72.7)	255 (72.2)	289 (73.2)	1274 (68.7)	
missing	0 (0)	0 (0)	0 (0)	0 (0)	0 (0)	
**Kinship to the deceased**						**<0.001**
blood-related	934 (84.5)	376 (50.3)	183 (51.8)	193 (48.9)	1310 (70.7)	
non-blood-related	168 (15.2)	369 (49.3)	170 (48.1)	199 (50.4)	537 (29.0)	
missing	4 (0.4)	3 (0.4)	0 (0)	3 (0.8)	7 (0.4)	

^†^*p*-values for bivariate associations in relation to main comparison (bereavement by sudden natural causes versus bereavement by any sudden unnatural causes), with those in bold below the threshold for significance (<0.05). ^††^ covariates included in adjusted model. ^a^ age and time since bereavement were used as continuous variables in our multivariable models but are presented here as binary variables for ease of interpretation. ^b^ socio-economic status using the five categories from UK Office for National Statistics. ^c^ measured using CIDI screen for depression.

**Table 2 ijerph-19-13245-t002:** Associations between bereavement by unnatural causes and perceived reduction/cessation in substance use.

Exposure to Bereavement by:	Sudden Natural Causes (n = 1106)		Sudden Unnatural Causes, Including Suicide (n = 748)					Total (n = 1854)
Outcome	Prevalence n (%)	Odds Ratio	Prevalence n (%)	Unadjusted Odds Ratio (95% CI)	*p*-Value ^†^	Adjusted ^a^ Odds Ratio (95% CI)	*p*-Value ^†^	Prevalence n (%)
Perceived reduction or cessation in alcohol use post-bereavement	78 (7.1)	1	64 (8.6)	1.21 (0.85–1.71)	0.287	1.24 (0.87–1.77)	0.239	142 (7.7)
Perceived reduction or cessation in drug use post-bereavement	19 (1.7)	1	31 (4.1)	2.63 (1.45–4.73)	**0.001**	2.61 (1.44–4.71)	**0.001**	50 (2.7)

^a^ adjusted for age, gender, socio-economic status, time since bereavement, pre-bereavement depression, and family history of psychiatric problems (including drug and alcohol problems). ^†^
*p*-values in bold are below the threshold for significance (<0.05).

**Table 3 ijerph-19-13245-t003:** Sensitivity analyses showing the associations between bereavement exposure sub-type and perceived reduction/cessation in substance use.

Exposure to Bereavement by:	Sudden Natural Causes (n = 1106)		Suicide (n = 353)					Total Sample for Sub-Analysis (n = 1501)
Outcomes	Prevalence n (%)	Odds Ratio	Prevalence n (%)	Unadjusted Odds Ratio (95% CI)	*p*-Value	Adjusted ^a^ Odds Ratio (95% CI)	*p*-Value	Prevalence n (%)
Perceived reduction or cessation in alcohol use post-bereavement	78 (7.1)	1	30 (8.5)	1.20 (0.77–1.87)	0.431	1.25 (0.80–1.97)	0.328	108 (7.4)
Perceived reduction or cessation in drug use post-bereavement	19 (1.7)	1	18 (5.1)	3.25 (1.76–6.32)	**0.001**	3.13 (1.60–6.13)	**0.001**	37 (2.5)
**Exposure to Bereavement by:**	**Sudden Natural Causes** **(n = 1106)**		**Sudden (Non-Suicide) Unnatural Causes** **(n = 395)**					**Total Sample for Sub-Analysis** **(n = 1459)**
**Outcomes**	**Prevalence** **n (%)**	**Odds Ratio**	**Prevalence** **n (%)**	**Unadjusted Odds Ratio** **(95% CI)**	***p*-Value ^†^**	**Adjusted** **^a^ Odds Ratio** **(95% CI)**	***p*-Value ^†^**	**Prevalence** **n (%)**
Perceived reduction or cessation in alcohol use post-bereavement	78 (7.1)	1	34 (8.6)	1.21 (0.79–1.86)	0.375	1.24 (0.80–1.92)	0.328	112 (7.5)
Perceived reduction or cessation in drug use post-bereavement	19 (1.7)	1	13 (3.3)	2.07 (1.01–4.28)	**0.048**	2.19 (1.06–4.53)	**0.035**	32 (2.1)

^a^ adjusted for age, gender, socio-economic status, time since bereavement, pre-bereavement depression, and family history of psychiatric problems (including drug and alcohol problems). ^†^
*p*-values in bold are below the threshold for significance (<0.05).

## Data Availability

Data are made available to researchers on formal application to the research team, subject to approval of an honorary contract at UCL.

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
