# Peer review of "Investigating Whether Bereavement by Suicide and Other Unnatural Causes Has a Deterrent Effect on Alcohol or Drug Use in Young Adults"

_ijerph, 2022, doi:10.3390/ijerph192013245_

Round 1
Reviewer 1 Report
This study highlights relevant evidence for both bereavement practitioners and substance use practitioners. This study appears to be original in examining bereavement types against substance use. There are weaknesses to the study but the authors highlight that findings are useful to provoke further research.
Taking a substance use practice perspective, there are biases in the cohort examined that could be acknowledged beyond the lack of power. Is the proportion of the sample by gender representative of bereaved young people (I suspect males are more likely to suffer unexpected and unnatural deaths than women, hence a higher proportion of bereaved women). also, the age range division needs explaining. is this bivariate created from the median? I don't think it is stated, but this is relevant as older age is a significant finding but the range is 22 - 40. This makes less societal sense and will be less sensitive to capturing intimate partnership loss. Another potential bias in the findings for reduced drug use among a predominantly female cohort is that women drug users are more likely than men to have a drug-using partner - who are also commonly controlling of the woman's drug use. See Arpa, 2017 and Moore & Measham, 2013 for example. Unnatural deaths among drug users are commonly overdoses, suicidal or not (often a subjective coronial decision and difficult to discern the difference), which could make a difference to the woman's relationship to drugs. It would be interesting also to see if another's fatal drug overdose is a motivational factor for the partner/close friend's reduced drug use too. The authors do address the point that mortality anxiety may be a motivational driver generally.
it is clear that the authors have performed analysis on the available data but it might be worth considering the potential biases inherent in including possible overdose deaths (which will be a higher proportion of deaths among young people), and the age group examined. The age of the cohort examined also needs to be included in the title to make it clearer to researchers interested in both young people and substance use.
Author Response
Reviewer 1
This study highlights relevant evidence for both bereavement practitioners and substance use practitioners. This study appears to be original in examining bereavement types against substance use. There are weaknesses to the study but the authors highlight that findings are useful to provoke further research.
Thank you for these comments, particularly noting where we need to acknowledge methodological weaknesses more clearly.
Taking a substance use practice perspective, there are biases in the cohort examined that could be acknowledged beyond the lack of power. Is the proportion of the sample by gender representative of bereaved young people (I suspect males are more likely to suffer unexpected and unnatural deaths than women, hence a higher proportion of bereaved women).
We agree that the gender balance in this survey sample is unlikely to be representative of young bereaved people. In our population-based analysis of Danish registry data to identify individuals bereaved by the death of a first degree relative (parents, offspring, spouses/partners, siblings), for bereavement by suicide, we found the male:female ratio of suicide-bereaved individuals to be 0.97, and for other bereavements we found the ratio to be 0.67 (Pitman et al, 2022 Proportion of suicides in Denmark attributable to bereavement by the suicide of a first-degree relative or partner: Nested case–control study. Acta Psychiatr Scand https://onlinelibrary.wiley.com/doi/10.1111/acps.1349 ). These ratios applied to all age groups, and in younger people we would expect an equal risk in males and females of parental or sibling bereavement but a greater risk of females being bereaved by male peer/partner death due to the greater risk of suicides and accidental deaths in young men (Pitman et al, 2012 https://www.thelancet.com/journals/lancet/article/PIIS0140-6736(12)60731-4/fulltext ).
We have added a line to our limitations to acknowledge power issues:
“Nevertheless, low event rates in our sample of 1,854 bereaved adults meant we had limited power for models.”
We have also edited our limitations section to acknowledge the gender balance:
“As females of all ages have a greater risk of bereavement (Pitman et al, 2022), we would expect an excess of bereaved females aged 40 and below, particularly given the much higher suicide and accidental death rate in males than females in this age group (Pitman et al, 2012). However, our findings from a predominantly female (81%), white (90%), high socio-economic status (60%) sample may not be generalisable to all young bereaved adults in the UK or internationally.”
Also, the age range division needs explaining. is this bivariate created from the median? I don't think it is stated, but this is relevant as older age is a significant finding but the range is 22 - 40. This makes less societal sense and will be less sensitive to capturing intimate partnership loss.
We presented age as a binary variable in the tables to aid interpretation, but used age as a continuous measure in our regression models. We used this age 21 cut-off in a HEI sample as it corresponded with the age at which most undergraduates complete their course, approximating to an undergraduate and post-graduate or staff distinction. A minority (15%) had lost a non-blood related close contact, which included friends as well as intimate partners. We have previously reported that the majority of those reporting the loss of a non-blood-related contact were those bereaved by the death of a friend (74%), whilst only 11% reported the death of a partner, 4% of an ex-partner and 12% a step/adoptive/in-law family member (Pitman et al, 2016 https://bmjopen.bmj.com/content/6/1/e009948 ). We have added mean age to Table 1 to provide clarity over age.
Another potential bias in the findings for reduced drug use among a predominantly female cohort is that women drug users are more likely than men to have a drug-using partner - who are also commonly controlling of the woman's drug use. See Arpa, 2017 and Moore & Measham, 2013 for example. Unnatural deaths among drug users are commonly overdoses, suicidal or not (often a subjective coronial decision and difficult to discern the difference), which could make a difference to the woman's relationship to drugs. It would be interesting also to see if another's fatal drug overdose is a motivational factor for the partner/close friend's reduced drug use too. The authors do address the point that mortality anxiety may be a motivational driver generally.
We did not have sufficiently granularity of data to identify cause of death within the non-suicide unnatural deaths group, but agree that these correlates are likely to be highly relevant. We read the report by Arpa 2017 and the Chapter by Moore & Measham, 2013 and have added this point to our section on findings in the context of other studies:
“Given the predominance of females in our sample, our findings for drug use may reflect the impact of losing a drug-using partner as well as their facilitative influence on drug use at home or in leisure spaces (Arpa 2017; Moore & Measham, 2013).
it is clear that the authors have performed analysis on the available data but it might be worth considering the potential biases inherent in including possible overdose deaths (which will be a higher proportion of deaths among young people), and the age group examined. The age of the cohort examined also needs to be included in the title to make it clearer to researchers interested in both young people and substance use.
We agree about these potential biases and have added to our limitations:
“Given that drug-related deaths of peers and partners are likely to be over-represented in this younger sample, our findings may not be generalisable to other age groups.”
We have also added “in young adults” to our title.
Reviewer 2 Report
Overall a quality paper. A few comments:
1) The background would be improved by either (a) a statement that this work is novel and the question has not been explored or (b) a summary of the literature focused on suicide bereavement and substance use.
2) The authors say that recruitment is described elsewhere, but it would be helpful to provide more detail in the present study as it is a stand-alone report.
3) The content analyses -- turned quantitative -- the item is vulnerable to low sensitivity since not all people who experienced a particular pattern might think to describe it in that way. The limitations might want to discuss how this could have impacted their results and null findings (e.g., can the direction of bias in the effect be predicted?)
4) Did the authors explore the between-group prevalence of the fuller coded options (before dichotomizing)?
Author Response
Reviewer 2
Overall a quality paper. A few comments:
Thank you for these comments, particularly noting where we need to clarify background methods and acknowledge methodological weaknesses more clearly.
1) The background would be improved by either (a) a statement that this work is novel and the question has not been explored or (b) a summary of the literature focused on suicide bereavement and substance use.
We have added to the introduction that this hypothesis has not yet been investigated, and this is also mentioned at the start of the section describing findings in the context of other studies. We have included in the introduction a summary of all the key quantitative (references 6-9) and qualitative (references 10-14) literature we know of comparing substance misuse after suicide bereavement, other bereavement and in non-bereaved controls. This highlights that both suicide bereavement and non-suicide bereavement tend to carry a higher risk of substance misuse than no bereavement, but that no differences are observed when comparing suicide bereavement to bereavement by other unnatural causes. The many differences between these two groups have been noted previously (reference 3) and point to the trauma experienced by both groups after unnatural loss.
2) The authors say that recruitment is described elsewhere, but it would be helpful to provide more detail in the present study as it is a stand-alone report.
We have expanded the paragraph explaining how this sample were recruited.
3) The content analyses -- turned quantitative -- the item is vulnerable to low sensitivity since not all people who experienced a particular pattern might think to describe it in that way. The limitations might want to discuss how this could have impacted their results and null findings (e.g., can the direction of bias in the effect be predicted?)
We have addressed the issue of bias in relation to efforts to improve the validity of coding but agree that we need to make the point that validity of coding may have introduced bias through over- or under-ascertainment of outcomes, particularly where social desirability bias or recall bias operated. In the case of validity of coding, we cannot be clear about the direction of bias, as reflexivity was intended to reduce any inductive bias. We have added the following line to our limitations:
“However, we acknowledge that subjectivity in coding, social desirability bias, and recall bias may have over-ascertained (particularly in the context of social desirability bias) or under-ascertained our outcomes, biasing our quantitative findings.”
4) Did the authors explore the between-group prevalence of the fuller coded options (before dichotomizing)?
In the introduction and discussion we have cited an earlier paper (Pitman et al, 2020 https://www.frontiersin.org/articles/10.3389/fpsyg.2020.01024/full ) in which we described the prevalence of each of the 13 categories by mode of bereavement (suicide bereavement, other sudden unnatural deaths, sudden natural deaths) and tested whether people bereaved by suicide and by sudden unnatural deaths are more likely to increase their use of alcohol or drugs. We have clarified the findings of that paper in much more detail in our section on findings in the context of other studies. The current paper extends that work by testing a novel hypothesis relating solely to reduction or cessation in use of alcohol or drugs.